# BIAS-RESILIENT NEURAL NETWORK

## ABSTRACT

Presence of bias and confounding effects is inarguably one of the most critical challenges in machine learning applications that has alluded to pivotal debates in the recent years. Such challenges range from spurious associations of confounding variables in medical studies to the bias of race in gender or face recognition systems. One solution is to enhance datasets and organize them such that they do not reflect biases, which is a cumbersome and intensive task. The alternative is to make use of available data and build models considering these biases. Traditional statistical methods apply straightforward techniques such as residualization or stratification to precomputed features to account for confounding variables. However, these techniques are generally not suitable for end-to-end deep learning methods. In this paper, we propose a method based on the adversarial training strategy to learn discriminative features unbiased and invariant to the confounder(s). This is enabled by incorporating a new adversarial loss function that encourages a vanished correlation between the bias and learned features. We apply our method to synthetic data, medical images, and a gender classification (Gender Shades Pilot Parliaments Benchmark) dataset. Our results show that the learned features by our method not only result in superior prediction performance but also are uncorrelated with the bias or confounder variables. The code is available at http://blinded_for_review/.

## 1 INTRODUCTION

A central challenge in practically all machine learning applications is the consideration of confounding biases. Confounders are extraneous variables that distort the relationship between the input (independent) and output (dependent) variables and hence lead to erroneous conclusions (Pourhoseingholi et al., 2012). In a variety of applications ranging from disease prediction to face recognition, where machine learning models are built to predict labels from images, demographic variables (such as age, sex, race) of the study may confound the training process if the distribution of image labels is skewed with respect to them. In this situation, the predictor may learn the influence of the confounder and bias present in the data instead of actual discriminative cues.

It is a cumbersome task to account for all biases when curating large-scale datasets (Yang et al., 2019). An alternative approach is to account for the bias in the model. Traditionally, confounding variables are often controlled by statistical methods in either design or analytical stages (Aschengrau & Seage, 2013). In the design stage, one can utilize randomization or matching of the confounding variables across different study groups. In the analytical stage, confounding can be controlled by standardization or stratification (Pourhoseingholi et al., 2012; Aschengrau & Seage, 2013).

Another common solution is to learn the influence of the confounding variables on the input (independent) variables by regression analysis. Then, the residuals derived from the optimal regression model are regarded as the confounder-free input to train the predictor (Wodtke, 2018). The regression analysis works reasonably well under the assumption that the input variables represent deterministic features that are comparable across a population, *e.g.*, morphometric measurements extracted from medical images or engineered features extracted from face images. The method fails, however, when this assumption does not hold such as for the pixel intensity values in images. Note, the raw intensities are only meaningful within a neighborhood but variant across images. Therefore, these regression approaches cannot be used in connection with deep learning methods that are di-

---

† indicates equal contribution.

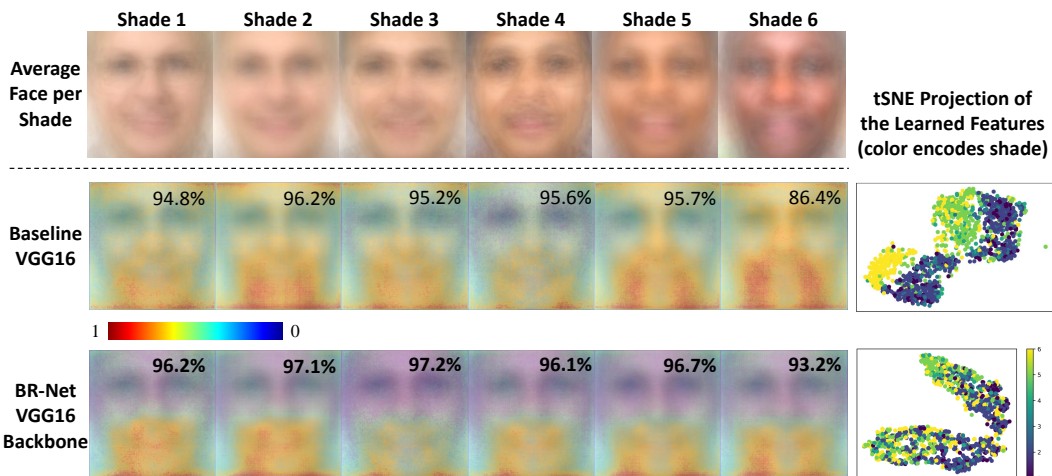

Figure 1: Average face images across each shade category (first row), average saliency map of the trained baseline (second row), and BR-Net (third row) color-coded with the normalized saliency value for each pixel. BR-Net results in more stable patterns across all 6 shade categories. The last column shows the tSNE projection of the learned features by each method. Our method results in a better feature space invariant to the bias variable (shade) while the baseline shows a clear pattern affected by the bias. Average accuracy of per-shade gender classification over 5 runs of 5-fold cross-validation is shown on each average map. The models are pre-trained on ImageNet and fine-tuned on GS-PPB. BR-Net is not only able to close the gap of accuracy for the darker shade but it also regularizes the model to improve per-category accuracy.

rectly applied to images, such as convolutional neural networks (CNNs). Removing confounding factors for CNNs is an open question we aim to address here.

We propose a feature learning scheme to produce features that are predictive of class labels while being unbiased to confounding variables. The idea is inspired by the domain-adversarial training approaches (Ganin et al., 2016) with controllable invariance (Xie et al., 2017) within the context of generative adversarial networks (GANs) (Goodfellow et al., 2014), but we argue that generic and widely used loss functions are not designed for controlling the invariance with respect to bias variables. Hence, we introduce an adversarial loss function that aims to quantify the statistical dependence between the learned features and bias variables with the correlation coefficient. This strategy improves over the commonly used cross-entropy or mean-squared error (MSE) loss that only aims to predict the exact value of the bias variables and thereby achieves stabler results within the context of adversarial training. Since our proposed model injects resilience towards the bias during training to produce confounder-invariant features, we refer to our approach as Bias-Resilient Neural Network (BR-Net).

We evaluate BR-Net on three datasets to examine different aspects of the method and compare it with a wide range of baselines. First, we test on a *synthetic dataset* to outline how the learned features by our method are unbiased to controlled confounding variables. Then, we test it on a *medical imaging application*, *i.e.*, predicting the human immunodeficiency virus (HIV) diagnosis directly from T1-weighted Magnetic Resonance Images (MRIs). As widely explored in the HIV literature, HIV disease accentuates brain aging (Cole et al., 2017) and if a predictor is learned not considering age as a confounder, the predictor may actually be learning the brain aging patterns rather than actual HIV markers. Lastly, we evaluate BR-Net for *gender classification* using the Gender Shades Pilot Parliaments Benchmark (GS-PPB) dataset (Buolamwini & Gebru, 2018). We use different backbones pre-trained on ImageNet (Deng et al., 2009) and fine-tune them for predicting gender from face images. We show that prediction of the vanilla models is dependent on the race of the subject (alternatively we consider skin color quantified by the 'shade' variable) and show poor results for darker faces, while BR-Net can successfully close the gap. Our comparison with methods based on multi-task (Lu et al., 2017) prediction (*i.e.*, predicting gender and shade as two tasks) and categorical GAN (Springenberg, 2015) (*i.e.*, predicting shade as a categorical variable in the adver-

sarial component) shows that BR-Net is not only able to learn features impartial to the bias of race (verified by feature embedding and saliency visualization), it also results in better performance in gender prediction (see Fig. 1).

## 2 RELATED WORK

**Fairness in Machine Learning:** In recent years, developing fair machine learning models have been the center of many discussions (Liu et al., 2018; Hashimoto et al., 2018; Barocas et al., 2017) even in the news outlets and media (Khullar, 2019; Miller, 2015). It has been argued that the bias essentially comes from human or society biases induced by the training datasets (Barocas & Selbst, 2016). Recent efforts in solving this problem have been focused on building fairer and more diverse datasets (Yang et al., 2019; Celis et al., 2016). However, this approach is not always practical for large-scale datasets or especially in medical applications, where data is relatively scarce and expensive to generate. In this work, we propose to use existing sets of data but to build models mindful of biases by learning impartial features that are only predictive of the actual output variables.

**Domain-Adversarial Training:** (Ganin et al., 2016) proposed for the first time to use adversarial training for domain adaptation tasks by creating a component in the network that uses the learned features to predict which domain the data is coming from (a binary variable; source or target). Ever since, several other works built on top of the same idea explored different loss functions (Bousmalis et al., 2017), domain discriminator settings (Tzeng et al., 2017), or cycle-consistency (Hoffman et al., 2017). The focus of all these works was to close the domain gap, which is often encoded as a binary variable. To learn generic bias-resilient models, we argue that we need to go beyond this and learn features that are invariant to either discrete or continuous confounders.

**Invariant Feature Learning:** There have been different attempts in the literature for learning representations that are invariant to specific factors in the data. For instance, (Zemel et al., 2013) took an information obfuscation approach to obfuscate membership in the protected group of data during training, and (Bechavod & Ligett, 2017; Ranzato et al., 2007) introduced a regularization-based method. Recently, (Xie et al., 2017; Akuzawa et al., 2019; Zhang et al., 2018; Elazar & Goldberg, 2018) proposed to use domain-adversarial training strategies for controllable invariant feature learning with respect to existing variables in the data. Some concurrent works (Sadeghi et al., 2019; Wang et al., 2019) have also used adversarial techniques for mitigating the effect of bias. These methods used similar adversarial loss functions as in domain adaptation that aim at predicting exact values of the bias variables. For instance, (Wang et al., 2019) used a binary cross-entropy for removing effect of 'gender' and (Sadeghi et al., 2019) used linear (and kernelized) least squares predictors as the adversarial component. Our study shows that these strategies fail at creating resilience against biases that take continuous or ordinal values. Instead, we introduce a loss function based on correlation coefficient to naturally alleviate the bias effects on the learned features.

**Distribution Matching:** Some previous work attempted to learn distributionally robust techniques to avoid learning confounded effects from data (Oren et al., 2019). This can be done by matching the distributions of the data (Cao et al., 2018; Baktashmotlagh et al., 2016) across different domains. However, distribution matching techniques only model data of a population as a whole and fall short when it is crucial to remove the association between the learned features and a specific bias or confounding variable for each single input data point. Whereas, to close the gap with respect to the underlying bias in the data, our correlation-based analysis minimizes the bias-predictive power of the learned features for every individual data point, which by construction harmonizes the data distribution on the population level.

## 3 BIAS-RESILIENT NEURAL NETWORK (BR-NET)

Suppose we have an $M$-class classification problem, for which we have $N$ pairs of training images and their corresponding target label(s): $\{(\mathbf{X}_i, \mathbf{y}_i)\}_{i=1}^N$. Assume that the study is confounded or biased by a set of $k$ variables, denoted by a vector $\mathbf{b} \in \mathbb{R}^k$. To train a deep neural network for classifying each image to its label(s) while not being biased by the confounders in the study, we propose our *end-to-end* architecture as in Fig. 2 similar to domain-adversarial training approaches (Ganin et al., 2016). Given the input image $\mathbf{X}$, we first apply a Feature Extraction ($\mathbb{FE}$) network, resulting in a feature vector $\mathbf{F}$. A Classifier ($\mathbb{C}$) is built on top of this feature vector to predict the

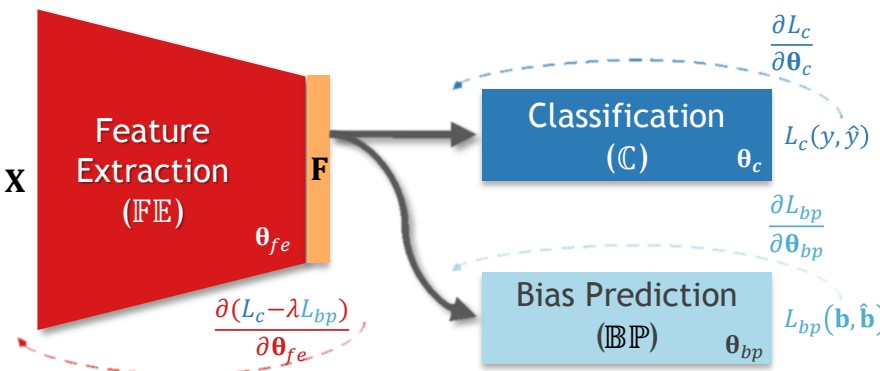

Figure 2: BR-Net architecture: $\mathbb{FE}$ learns features, $\mathbf{F}$, that successfully classify ($\mathbb{C}$) the input while being invariant (not correlated) to the bias variable(s), $\mathbf{b}$, using $\mathbb{BP}$ and the adversarial loss component, $-\lambda L_{bp}$, which is based on correlation coefficient. Forward arrows show forward paths while the backward dashed ones indicate back-propagation with the respective gradient values.

class label $\mathbf{y}$ for the input $\mathbf{X}$, and it forces $\mathbb{FE}$ to learn discriminative futures for the classification task. Now, to guarantee that these features are not biased to $\mathbf{b}$, we build another network (denoted by $\mathbb{BP}$) with a new loss function for predicting the bias variables from $\mathbf{F}$. We propose to back-propagate this loss to the feature extraction module in an adversarial way. As a result, the feature extractor learns features that minimize the classification loss, while maximizing the bias predictor loss.

Each network has its underlying trainable parameters, defined as $\boldsymbol{\theta}_{fe}$ for $\mathbb{FE}$, $\boldsymbol{\theta}_c$ for $\mathbb{C}$, and $\boldsymbol{\theta}_{bp}$ for $\mathbb{BP}$. If the predicted probability that subject $i$ belongs to class $m$ is defined by $\hat{y}_{im} = \mathbb{C}(\mathbb{FE}(\mathbf{X}_i; \boldsymbol{\theta}_{fe}); \boldsymbol{\theta}_c)$, the classification loss can be characterized by a cross-entropy:

$$L_c(\mathbf{X}, \mathbf{y}; \boldsymbol{\theta}_{fe}, \boldsymbol{\theta}_c) = -\sum_{i=1}^{N} \sum_{m=1}^{M} y_{im} \log(\hat{y}_{im}). \tag{1}$$

Similarly, with $\hat{\mathbf{b}}_i = \mathbb{BP}(\mathbb{FE}(\mathbf{X}_i; \boldsymbol{\theta}_{fe}); \boldsymbol{\theta}_{bp})$, we can define the adversarial component of the loss function. Standard methods for designing this loss function suggest to use a cross-entropy for binary/categorical variables (*e.g.*, in (Ganin et al., 2016; Xie et al., 2017)) or an $\ell_2$ MSE loss for continuous variables. However, we argue that in the context of bias control, the ultimate goal of adversarial training is to remove statistical association with respect to the bias variables, as opposed to maximizing the prediction error of them. In fact, the adversarial training based on MSE leads to the maximization of the $\ell_2$ distance between $\hat{\mathbf{b}}$ and $\mathbf{b}$, which could be trivially achieved by uniformly shifting the magnitude of $\hat{\mathbf{b}}$, thereby potentially resulting in an ill-posed optimization and oscillation in the adversarial training. To address this issue, we define the following surrogate loss for predicting the bias confounders while quantifying the statistical dependence with respect to $\mathbf{b}$:

$$L_{bp}(\mathbf{X}, \mathbf{b}; \boldsymbol{\theta}_{fe}, \boldsymbol{\theta}_{bp}) = -\sum_{\kappa=1}^{k} \text{corr}^2(\mathbf{b}_\kappa, \hat{\mathbf{b}}_\kappa), \tag{2}$$

where $\text{corr}^2(\cdot, \cdot)$ is the squared Pearson correlation between its inputs and $\mathbf{b}_\kappa$ defines the vector of $\kappa^{\text{th}}$ bias variable across all inputs. Through adversarial training, we aim to remove statistical dependence by encouraging a zero correlation between $\mathbf{b}_\kappa$ and $\hat{\mathbf{b}}_\kappa$. Note, $\mathbb{BP}$ deems to maximize squared correlation and $\mathbb{FE}$ minimizes for it; Since $\text{corr}^2$ is bounded in the range [0, 1], both minimization and maximization schemes are deemed feasible. Having these loss functions defined, the overall objective of the network is then defined as

$$\min_{\boldsymbol{\theta}_{fe}, \boldsymbol{\theta}_c} \max_{\boldsymbol{\theta}_{bp}} L_c(\mathbf{X}, \mathbf{y}; \boldsymbol{\theta}_{fe}, \boldsymbol{\theta}_c) - \lambda L_{bp}(\mathbf{X}, \mathbf{b}; \boldsymbol{\theta}_{fe}, \boldsymbol{\theta}_{bp}). \tag{3}$$

where hyperparameter $\lambda$ controls the trade-off between the two objectives.

This scheme is similar to GAN (Goodfellow et al., 2014) and domain-adversarial training (Ganin et al., 2016; Xie et al., 2017), in which a min-max game is portrayed between two networks. In our

case, $\mathbb{FE}$ extracts features that minimize the classification criterion, while fooling $\mathbb{BP}$ (*i.e.*, making $\mathbb{BP}$ incapable of predicting the bias variables). Hence, the saddle point for this optimization objective is obtained when the parameters $\theta_{fe}$ minimize the classification loss while maximizing the loss of the bias prediction module. Simultaneously, $\theta_c$ and $\theta_{bp}$ minimize their respective network losses.

## 3.1 THEORETICAL PROPERTY

In general, a zero-correlation or a zero-covariance only quantifies linear independence between variables but cannot infer non-linear relationships. However, we now theoretically show that, under certain assumptions on the adversarial training of $\mathbb{BP}$, a zero-covariance would guarantee the *mean independence* (Wooldridge, 2010) between bias variables and features, a much stronger type of statistical independence than the linear type.

A random variable $\mathcal{B}$ is said to be *mean independent* of $\mathcal{F}$ if and only if $E[\mathcal{B}|\mathcal{F} = \xi] = E[\mathcal{B}]$ for all $\xi$ with non-zero probability, where $E[\cdot]$ defines the expected value. In other words, the expected value of $\mathcal{B}$ is neither linearly nor non-linearly dependent on $\mathcal{F}$, but the variance of $\mathcal{B}$ might. The following theorem then relates the mean independence between features $\mathcal{F}$ and bias variables $\mathcal{B}$ to the zero-covariance between $\mathcal{B}$ and the prediction of $\hat{\mathcal{B}}$ produced by the adversarial component, $\mathbb{BP}$.

*Property 1: $\mathcal{B}$ is mean independent of $\hat{\mathcal{B}} \Rightarrow Cov(\mathcal{B}, \hat{\mathcal{B}}) = 0$.*

*Property 2: $\mathcal{B}, \mathcal{F}$ are mean independent $\Rightarrow \mathcal{B}$ is mean independent of $\hat{\mathcal{B}} = \phi(\mathcal{F})$ for any mapping function $\phi$.*

**Theorem 1.** *Given random variables $\mathcal{F}, \mathcal{B}, \hat{\mathcal{B}}$ with finite second moment,*

*$\mathcal{B}$ is mean independent of $\mathcal{F} \Leftrightarrow$ for any arbitrary mapping $\phi$, s.t. $\hat{\mathcal{B}} = \phi(\mathcal{F})$, $cov(\mathcal{B}, \hat{\mathcal{B}}) = 0$*

*Proof.* The forward direction $\Rightarrow$ follows directly through *Property* 1 and 2. We focus the proof on the reverse direction. Now, construct a mapping function $\hat{\mathcal{B}} = \phi(\mathcal{F}) = E[\mathcal{B}|\mathcal{F}]$, *i.e.*, $\phi(\xi) = E[\mathcal{B}|\mathcal{F} = \xi]$, then $Cov(\mathcal{B}, \hat{\mathcal{B}}) = 0$ implies

$$E\big[\mathcal{B}E[\mathcal{B}|\mathcal{F}]\big] = E[\mathcal{B}]E\big[E[\mathcal{B}|\mathcal{F}]\big]. \tag{4}$$

Due to the self-adjointness of the mapping $\mathcal{B} \mapsto E[\mathcal{B}|\mathcal{F}]$, the left hand side of Eq. (4) reads $E\big[\mathcal{B}E[\mathcal{B}|\mathcal{F}]\big] = E\big[\left(E[\mathcal{B}|\mathcal{F}]\right)^2\big] = E[\hat{\mathcal{B}}^2]$. By the law of total expectation $E\big[E[\mathcal{B}|\mathcal{F}]\big] = E[\mathcal{B}]$, the right hand side of Eq. (4) becomes $E[\hat{\mathcal{B}}]^2$. By Jensen's (in)equality, $E[\hat{\mathcal{B}}^2] = E[\hat{\mathcal{B}}]^2$ holds if and only if $\hat{\mathcal{B}}$ is a constant, *i.e.*, by definition, $\mathcal{B}$ is mean independent of $\mathcal{F}$.

$\square$

**Remark.** *In practice we normalize the covariance by standard deviations of variables for optimization stability. In the unlikely singular case that $\mathbb{BP}$ outputs a constant prediction, we add a small perturbation in computing the standard deviation.*

This theorem echoes the validity of our adversarial training strategy: $\mathbb{FE}$ encourages a zero-correlation between $\mathbf{b}_\kappa$ and $\hat{\mathbf{b}}_\kappa$, which enforces $\mathbf{b}_\kappa$ to be mean independent of $\mathbf{F}$ (one cannot infer the expected value of $\mathbf{b}_\kappa$ from $\mathbf{F}$). In turn, assuming $\mathbb{BP}$ has the capacity to approximate any arbitrary mapping function, the mean independence between features and bias would correspond to a zero-correlation between $\mathbf{b}_\kappa$ and $\hat{\mathbf{b}}_\kappa$, otherwise $\mathbb{BP}$ would adversarially optimize for a mapping function that increases the correlation.

## 3.2 IMPLEMENTATION DETAILS

Similar to the training of GANs, in each iteration, we first back-propagate the $L_c$ loss to update $\theta_{fe}$ and $\theta_c$. With $\theta_{fe}$ fixed, we then minimize the $L_{bp}$ loss to update $\theta_{bp}$. Finally, with $\theta_{bp}$ fixed, we maximize the $L_{bp}$ loss to update $\theta_{fe}$. The last step can be considered as the bias effect removal component. Furthermore, in the present study, $L_{bp}$ depends on the correlation operation, which is a population-based operation, as opposed to individual-level error metrics such as cross-entropy or MSE losses. Therefore, we calculate the correlations over each training batch as a batch-level

operation. Depending on the application, we can use different architectures for each of the three subnetworks. We use a 3D CNN (Esmaeilzadeh et al., 2018; Nie et al., 2016) for $\mathbb{FE}$ to extract features from 3D medical images and use VGG16 (Simonyan & Zisserman, 2015) and ResNet50 (He et al., 2015) backbones for GS-PPB. For $\mathbb{C}$ and $\mathbb{BP}$, we use a two-layer fully connected network.

## 4 EXPERIMENTS

In this section, we evaluate our method on three different scenarios. First, we run a synthetic experiment to verify the validity of our assumptions. Then, we apply BR-Net to predict diagnosis of HIV from brain MRIs confounded by the subjects' age. Finally, we test the model for predicting gender from face images and show how controlling for variables related to race (*e.g.*, face color shade) can robustly enhance prediction performance. We compare BR-Net with several baselines, and evaluate how the features learned by our method are invariant to the bias or confounding variables.

**Baseline Methods**. In line with the implementation of our approach, the baseline for all three experiments is a vanilla CNN, whose architecture is exactly the same as BR-net except that there is no bias prediction sub-network and hence the adversarial loss. We emphasize that BR-Net aims to remove the association between prediction and bias by encouraging vanished correlation, which is different from simply maximizing the prediction loss (w.r.t bias) as usually performed in many GAN settings. Therefore, the second comparison method is a BR-Net with the adversarial loss being the MSE, denoted by 'BR-Net (w/ MSE).' For the Gender Shades PPB experiment, we further add two other baseline methods, one predicting both 'gender' and 'shade' in a multi-task setting (Lu et al., 2017), denoted by 'Multi-Task'; and one replacing correlation loss function $L_{bp}$ with a cross-entropy loss as the 'shade' variable has a ordinal but categorical value. The adversarial training then relies on maximizing the entropy of $\mathbb{BP}$ predictions as motivated in Categoral GAN models ('CatGAN') (Springenberg, 2015). These baselines show how the correlation loss plays an important role in delineating the bias and confounding effects.

### 4.1 SYNTHETIC DATA

We generate a synthetic dataset comprised of two groups of data, each containing 512 images of resolution $32 \times 32$ pixels. Each image is generated by 4 Gaussians (see Fig. 3a), the magnitude of which is controlled by $\sigma_A$ and $\sigma_B$. For each image from Group 1, we sample $\sigma_A$ and $\sigma_B$ from a uniform distribution $\mathcal{U}(1, 4)$ while we generate images of Group 2 with stronger intensities by sampling from $\mathcal{U}(3, 6)$. Gaussian noise is added to the images with standard deviation $0.01$. Now we assume the difference in $\sigma_A$ between the two groups is associated with the true discriminative cues that should be learned by a classifier, whereas $\sigma_B$ is a given confounder. In other words, an unbiased model should predict the group label purely based on the two diagonal Gaussians and not dependent on the two off-diagonal ones. To show that the BR-Net can result in such models by controlling for $\sigma_B$, we train it on the whole dataset of 1,024 images given their respective binary labels and confounder values $\sigma_B$.

For simplicity, we construct the $\mathbb{FE}$ Network with 3 stacks of $2 \times 2$ convolution/ReLU/max-pooling layers to produce 32 features. Both the $\mathbb{BP}$ and $\mathbb{C}$ networks have one hidden layer of dimension 16 with $tanh$ as the non-linear activation function. After training, BR-Net achieves $89\%$ training accuracy and BR-Net w/ MSE achieves $90\%$. Note that the theoretically maximum training accuracy is $90\%$ due to the overlapping sampling range of $\sigma_A$ between the two groups. The baseline model, however, achieves $95\%$ accuracy, indicating that the model additionally relies on the confounding effects $\sigma_B$ for predicting the group label, an undesired behavior. To further investigate the association between the learned features and $\sigma_B$, we measure their squared distance correlation ($dcor^2$) (Székely et al., 2007) for the training samples in Group 1 (when there is no association between $\sigma_B$ and prediction, $dcor$=0 for either group). Distance correlation is a widely-used measure of dependence between two paired vectors of arbitrary dimensions. Fig. 3b shows that our method successfully removes the statistical association w.r.t $\sigma_B$ as the distance correlation drops dramatically with training iterations. On the other hand, the baseline model without the $\mathbb{BP}$ component learns features that constantly yield high correlation. Note that the adversarial loss based on MSE yields unstable $dcor^2$ measures potentially due to the ill-posed optimization of maximizing $\ell_2$ distance. Finally, the above results are further supported by the 2D tSNE (Maaten & Hinton, 2008) projection of the learned features as shown in Fig. 3c. The feature space learned by the baseline

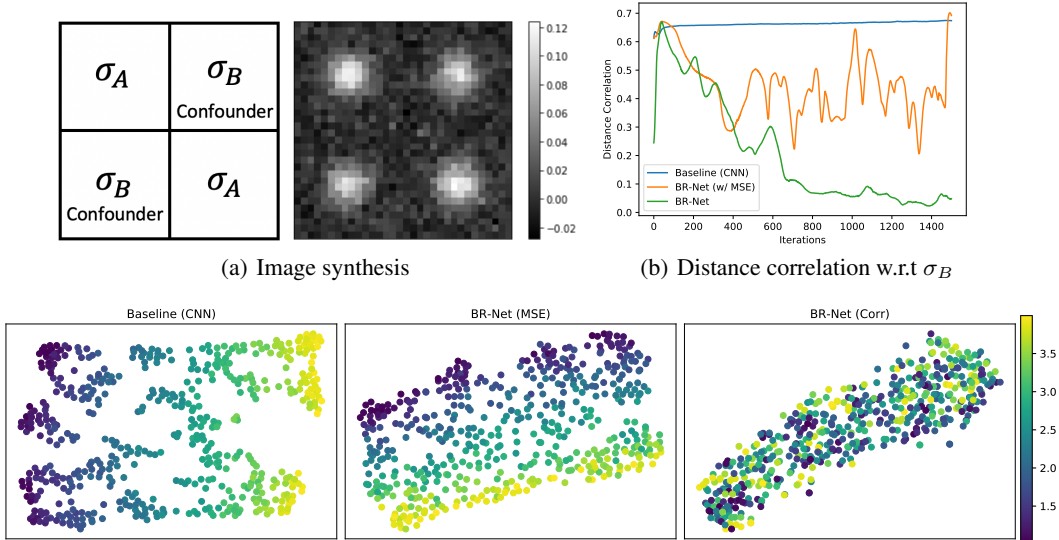

(a) Image synthesis       (b) Distance correlation w.r.t $\sigma_B$

(c) tSNE projection of the learned features for CNN (left), BR-Net w/ MSE (middle), and BR-Net (right)

Figure 3: Comparison of results on the synthetic dataset. Color indicates the value of $\sigma_B$.

Table 1: Balanced accuracy, $\mathbf{F}_1$-score, and area under curve (AUC) of HIV diagnosis prediction. Best results in each column are typeset in bold.

| Method | bAcc | $\mathbf{F}_1$ | AUC |
|---|---|---|---|
| Resid+SVM | 69.5 | 0.65 | 71.2 |
| Baseline (3D CNN) | 71.8 | 0.64 | 80.8 |
| BR-Net (w/ MSE) | 64.8 | 0.58 | 75.2 |
| BR-Net (Ours) | **74.2** | **0.67** | **80.9** |

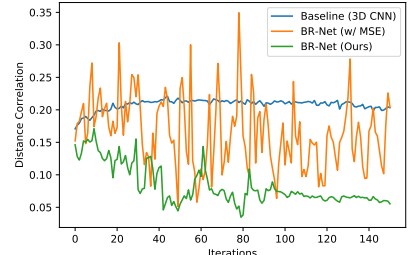

Figure 4: Distance correlation between the learned features and age for the CTRL cohort.

model forms a clear correlation with $\sigma_B$, whereas our method results in a space with no apparent bias. This confirms that the proposed adversarial technique successfully removes the bias from the confounding variable.

## 4.2 PREDICTION OF HIV DIAGNOSIS BASED ON MEDICAL IMAGES

Our second task aims at predicting the diagnosis of HIV patients *vs.* control subjects (CTRL) based on brain MRIs. The study cohort includes 223 CTRLs and 122 HIV patients who are seropositive for the HIV-infection with CD4 count $> 100 \frac{\text{cells}}{\mu L}$ (average: 303.0). Since the HIV subjects are significantly older in age than the CTRLs (CTRL: $45 \pm 17$, HIV: $51 \pm 8.3$, $p < .001$) in this study, age becomes a potential confounder; prediction of diagnosis labels may be dependent on subjects' age instead of true HIV markers. The T1-weighted MRIs are all skull stripped, affinely registered to a common template, and resized into a $64 \times 64 \times 64$ volume. Classification accuracy is measured with 5-fold cross validation. For each run, the training folds are augmented by random shifting (within one-voxel distance), rotation (within one degree) in all 3 directions, and left-right flipping based on the assumption that HIV infection affects the brain bilaterally (Adeli et al., 2018). The data augmentation results in a balanced training set of 1024 CTRLs and 1024 HIVs. As the flipping removes left-right orientation, the ConvNet is built on half of the 3D volume containing one hemisphere. The feature extractor $\mathbb{FE}$ has 4 stacks of $2 \times 2 \times 2$ 3D convolution/ReLu/batch-normalization/max-pooling layers yielding 4096 intermediate features. Both $\mathbb{BP}$ and $\mathbb{C}$ have one hidden layer of dimension 128 with $tanh$ as the activation function. For this experiment, as suggested in the previous work (Rao et al., 2017), confounding effects can only be reliably estimated among healthy subjects. So, in practice we only perform the adversarial loss back-propagation step for the CTRL group.

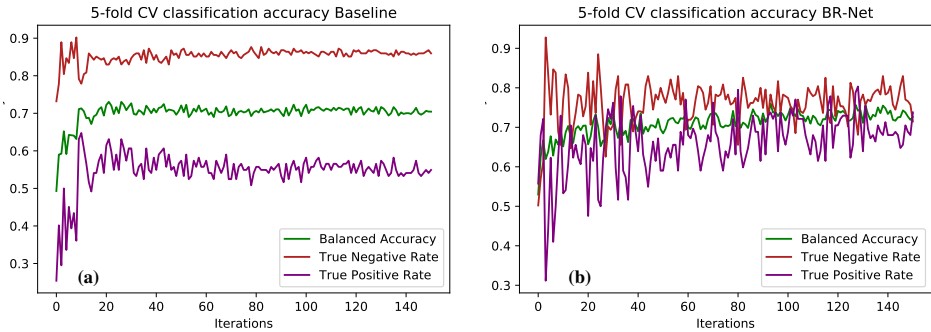

Figure 5: Accuracy, true negative, and true positive rates of the HIV diagnosis experiment, as a function of the number of iterations for (a) the 3D CNN baseline, (b) BR-Net. The results show that our adversarial training with a correlation loss function is robust against the imbalanced age distribution between HIV and CTRL subjects and achieves balanced prediction for both cohorts.

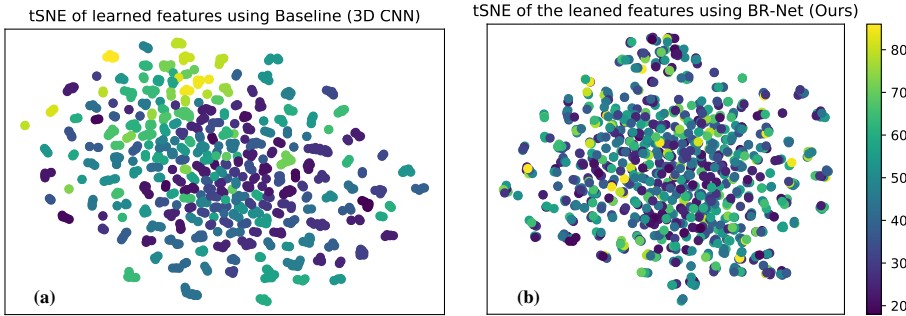

Figure 6: tSNE visualization of the learned features by (a) the 3D CNN baseline and (b) our BR-Net. Each point shows a subject in the CTRL cohort color-coded by their age.

Table 1 shows the diagnosis prediction accuracy of BR-Net in comparison with 3D CNN, BR-Net (w/ MSE), and Resid+SVM (note, to compare with the traditional residualization methods, we extract 298 brain regional measurements, residualize the confounders using a general linear model, and classify with a support vector machine). Our method (BR-Net) results in the most accurate prediction in terms of balanced accuracy (bAcc), area under curve (AUC), and $F_1$-score from the cross-validation. These results show that our method is able to learn discriminative features while controlling for confounders. In addition, we record the balanced accuracy, true positive, and true negative rate for each training iteration. As shown in Fig. 5, the baseline tends to predict most subjects as CTRLs (high true negative rate). This is potentially caused by the CTRL group having a wider age distribution, so an age-dependent predictor would bias the prediction towards CTRL. On the other hand, when controlling age as a confounder, BR-Net reliably results in balanced true positive and true negative rates.

Similar to the previous experiment, we train different methods on the entire dataset and plot the squared distance correlation between the learned features and confounders for the CTRL cohort (Fig. 4). The figure shows that for BR-Net the distance correlation between the features and the confounding variable (age) decreases with the adversarial training. Whereas, the baseline model 3D CNN consistently produces features that are highly correlated with the confounder, and BR-Net w/ MSE produces inconsistent and unreliable associations with respect to the confounder. The t-SNE (Maaten & Hinton, 2008) projection of the learned feature spaces are visualized in Fig. 6. The feature space learned by the baseline model forms a clear association with age, as older subjects are concentrated on the top left region of the space. This again suggests predictions from the baseline may be dependent on age rather than true HIV markers. Whereas, our method results in a space with no apparent bias to age.

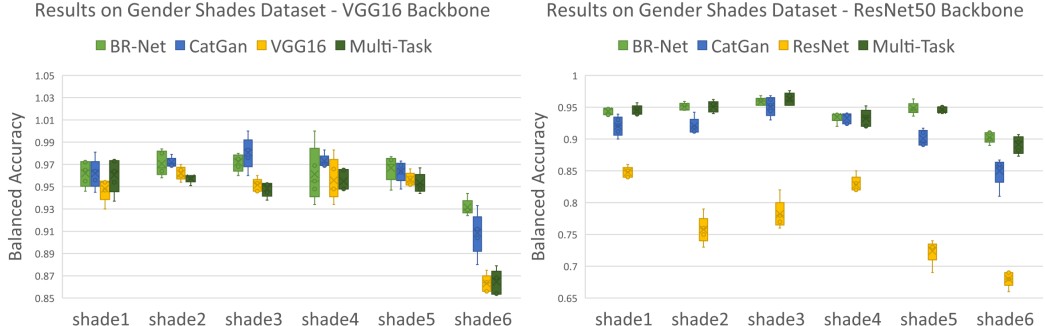

Figure 7: Accuracy of gender prediction from face images across all shades (1 to 6) of the GS-PPB dataset with two backbones, (left) VGG16 and (right) ResNet50. BR-Net consistently results in more accurate predictions in all 6 shade categories by injecting bias-resilience into the model.

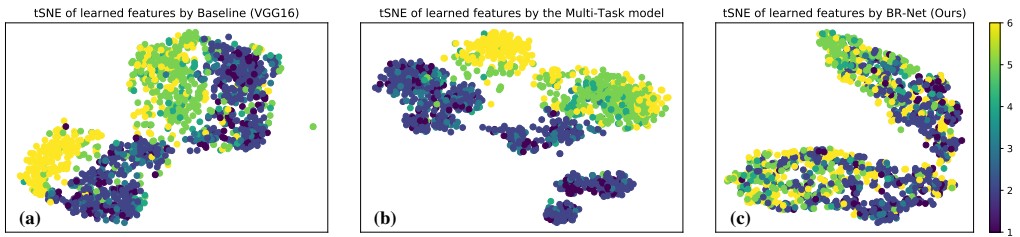

Figure 8: tSNE visualization of the learned features by (a) the VGG16 baseline, (b) a multi-task baseline, and (c) our BR-Net. Each point shows an image in the dataset color-coded with 'shade'.

## 4.3 GENDER PREDICTION USING THE GS-PPB DATASET

The last experiment is on gender prediction from face images in the Gender Shades Pilot Parliaments Benchmark (GS-PPB) dataset (Buolamwini & Gebru, 2018). This dataset contains 1,253 facial images of 561 female and 692 male subjects. The face shade is quantified by the Fitzpatricksix-point labeling system and is categorised from type 1 (lighter) to type 6 (darker). This quantization was used by dermatologistsuse for skin classification and determining risk for skin cancer (Buolamwini & Gebru, 2018).

To train our models on this dataset, we use backbones VGG16 (Simonyan & Zisserman, 2015) and ResNet50 (He et al., 2015) pre-trained on ImageNet (Deng et al., 2009). We fine-tune each model on GS-PPB dataset to predict the gender of subjects based on their face images using fair 5-fold cross-validation. The ImageNet dataset for pre-training the models has fewer cases of humans with darker faces (Yang et al., 2019), and hence the resulting models have an underlying bias to the shade.

BR-Net counts the variable 'shade' as an ordinal and categorical bias variable. As discussed earlier, besides the vanilla VGG16 and ResNet50 models, we compare the results with a multi-task baseline (Lu et al., 2017), which predicts both 'gender' and 'shade' simultaneously, and a model that uses the entropy loss as the adversarial loss for the cross-entropy-based categorical prediction (proposed by CatGAN (Springenberg, 2015)). Table 2 shows the results across five runs of 5-fold cross-validation. Fig. 7 plots the accuracy for each individual 'shade' category. As can be seen from the table and

Table 2: Average results over five runs of 5-fold cross-validation on the GS-PPB dataset. Best results in each column are typeset in bold.

| Method | VGG16 Backbone | | | ResNet50 Backbone | | |
|---|---|---|---|---|---|---|
| | bAcc | $F_1$ | AUC | bAcc | $F_1$ | AUC |
| Baseline | $94.1 \pm 0.2$ | $93.5 \pm 0.3$ | $98.9 \pm 0.1$ | $75.7 \pm 2.0$ | $68.0 \pm 3.0$ | $96.2 \pm 0.3$ |
| CatGAN | $96.0 \pm 0.5$ | $95.7 \pm 0.5$ | $\mathbf{99.4 \pm 0.2}$ | $90.1 \pm 1.0$ | $90.0 \pm 1.0$ | $96.3 \pm 0.7$ |
| Multi-Task | $94.0 \pm 0.3$ | $93.4 \pm 0.3$ | $98.9 \pm 0.1$ | $94.0 \pm 0.3$ | $93.4 \pm 0.3$ | $98.6 \pm 0.3$ |
| BR-Net | $\mathbf{96.3 \pm 0.6}$ | $\mathbf{96.0 \pm 0.7}$ | $\mathbf{99.4 \pm 0.2}$ | $\mathbf{94.1 \pm 0.2}$ | $\mathbf{93.6 \pm 0.2}$ | $\mathbf{98.6 \pm 0.1}$ |

the figure, BR-Net outperforms other methods on average while producing similar accuracy across all 'shade' categories. Prediction made by other methods, however, is more dependent on the bias variable by showing inconsistent recognition capabilities for different 'shade' categories and failing significantly on darker faces. This bias is confirmed by the tSNE projection of the feature spaces learned by different methods (see Fig. 8). The features learned by the vanilla baseline or even the multi-task model show a clear dependency on the 'shade' while BR-Net results in a roughly uniform distribution of subjects.

To gain more insight into the results, we visualize the saliency maps derived for the baseline and BR-Net. For this purpose, we use a similar technique as in (Simonyan et al., 2014) to extract the pixels in the original image space highlighting the areas that are discriminative for the gender labels. Generating such saliency maps for all inputs, we visualize the average map for each individual 'shade' category (Fig. 1). The value on each pixel corresponds to the attention from the network to that pixel within the classification process. Compared to the baseline, BR-Net focuses more on specific face regions and results in more stable patterns across all 'shade' categories.

## 5 CONCLUSION

We proposed a method based on adversarial training strategies by encouraging vanished correlation to learn features for the prediction task while being unbiased to the confounding variables in the study. We evaluated our bias-resilient neural network (BR-Net) on a synthetic, a medical diagnosis, and a gender prediction dataset. In all experiments, BR-Net resulted in a feature embedding space that was agnostic to the bias in the data while all other methods failed to do so. Based on our experiments we can conclude that, besides the attempt to improve datasets and curate unbiased ones (Yang et al., 2019), it is crucial to build models that properly account for the bias in data during training. Our bias-resilient model and some other recent works set on foot toward this direction. This is crucial as machine learning models are acceding to everyday lives, or are being developed for crucial medical applications. Failure to account for the underlying bias or confounding effects can lead to spurious associations and erroneous decisions. As a direction for the future work, other strategies such as deep canonical correlation analysis (Andrew et al., 2013) can be explored to form the adversarial component.

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
