# OpenReview forum: "Bias-Resilient Neural Network"
_ICLR.cc/2020/Conference — Reject_

### Official Review · AnonReviewer3 · 2019-10-23
**Official Blind Review #3**

**Rating:** 8

**Review:**

This paper presents Bias-Resilient neural network (BR-Net) that is designed to learn representations that can accurately predict the desired target while being invariant to the confounding covariates in the data. The proposed method is based on domain adversarial training strategies, especially that of (Ganin et al., 2016), where the adversarial component is modified from “loss of distinguishing between the source and target domains” to “the squared Pearson correlation between the ground truth bias covariate and its estimation from the learned representation”. This design is based on the argument that the ultimate goal of bias control here is removing statistical association with respect to the bias variables, as opposed to maximizing the prediction error of them.

Things to improve the paper that did not impact the score:
	- Introduction, line 4: Wrong citation format: use of \citet instead of \citep. Correct for all citations throughout the paper.

References:
	- Ganin, Y., Ustinova, E., Ajakan, H., Germain, P., Larochelle, H., Laviolette, F., ... & Lempitsky, V. (2016). Domain-adversarial training of neural networks. The Journal of Machine Learning Research, 17(1), 2096-2030.


**Experience Assessment:**

I have read many papers in this area.

**Review Assessment: Checking Correctness Of Derivations And Theory:**

I assessed the sensibility of the derivations and theory.

**Review Assessment: Checking Correctness Of Experiments:**

I assessed the sensibility of the experiments.

**Review Assessment: Thoroughness In Paper Reading:**

I read the paper at least twice and used my best judgement in assessing the paper.

---

> ### Author Response · Authors · 2019-11-07
> **Response to AnonReviewer3**
>
> >>> Comment: This paper presents Bias-Resilient neural network (BR-Net) that is designed to learn representations that can accurately predict the desired target while being invariant to the confounding covariates in the data. The proposed method is based on domain adversarial training strategies, especially that of (Ganin et al., 2016), where the adversarial component is modified from “loss of distinguishing between the source and target domains” to “the squared Pearson correlation between the ground truth bias covariate and its estimation from the learned representation”. This design is based on the argument that the ultimate goal of bias control here is removing statistical association with respect to the bias variables, as opposed to maximizing the prediction error of them.
> References:
> - Ganin, Y., Ustinova, E., Ajakan, H., Germain, P., Larochelle, H., Laviolette, F., ... & Lempitsky, V. (2016). Domain-adversarial training of neural networks. The Journal of Machine Learning Research, 17(1), 2096-2030.
>
> > Response: We appreciate the positive comments from this reviewer. We agree that this is a very important problem that needs to be addressed for deep learning models that has been neglected for long, especially for medical applications.
>
> ————-
> >>> Comment: Things to improve the paper that did not impact the score:
> - Introduction, line 4: Wrong citation format: use of \citet instead of \citep. Correct for all citations throughout the paper.
>
> > Response: Thanks for pointing this out. We have fixed the citation formatting.

---

### Official Review · AnonReviewer1 · 2019-10-23
**Official Blind Review #1**

**Rating:** 1

**Review:**

The authors propose a method based on GAN  to classify data while automatically removing confounding effects during training, in order to obtain a classifier whose features are not biased by any confounding effect. The proposed idea is based on the extension of classical classification architectures to account for bias prediction. In practice, the parameters of the feature extraction component are updated to solve both bias prediction and the desired classification problem in adversarial fashion. Pearson’s correlation was chosen as a metric for bias.

The paper is interesting and addresses an important problem for the application of machine learning methods in several real context. The feeling is however that the paper should have better explored the implication of the proposed model of bias, and better investigated the relationship with simpler approaches relying on similar hypothesis.

Here are my main comments for this work:

- Why Pearson’s correlation should be a reliable metric to quantify bias? This metric is insensitive to affine scaling of the data, which is a quite common form of bias (for example in medical images).
- The authors should have investigated the relationship between the proposed method and bias removal through canonical correlation analysis (CCA), and perhaps its non-linear variants. At the end this is what their network is doing, although in an end-to-end fashion. Using the CCA projections in the latent space for classification would be the closest approach to the state of the art for bias removal in statistical analysis (residual analysis).
- The experimental setting illustrated in 4.1 is not clear. In which sense \sigma_A is a common factor for the two groups? Why the theoretical maximum classification accuracy is 90%? Figure 2 is not clear either and doesn’t help understanding the structure of the generated data (e.g. axis labels missing, colorbars units not specified).
- It is not clear why authors quantify the correlation in the latent space with tSNE projections. tSNE is highly sensitive to the choice of parameters and it would be important to ensure that it was a “fair competition” between all the methods, when showing the results of the dimensionality reduction. This is another modelling step relying on specific assumptions which decreases interpretability of the findings. The authors also proposed to assess the decorrelation of the estimated features throughout the different methods by measuring the squared distance correlation. Naturally, their method is the one which exhibits the best performances using this metric. However, this way of assessing the decorrelation of the features with the biases is unfair to the other methods, as in their case they specifically built their model to avoid statistical correlation between features and biases.
- Experiment 4.2 has some controversial aspects, as the bias correction is performed on the control population only, while the model is trained on the entire population. I understand the fact that confounding can be estimated only on healthy conditions, however in this case the network is going to be biased by the control group by construction. The effect of such a choice in the end-to-end optimisation scheme is really not clear.
- We also observe that the results of the baseline CNN are very close to the BR-Net. The main difference lies in the fact that the CNN tends to have an unbalanced classification between true negatives and true positives. However, what would happen if we corrected for age before applying the CNN ?
- In the case where the performances of the CNN would be improved, this last question would raise another one. Indeed, if I already know the confounding effects I want to correct for, why wouldn’t I correct them beforehand in order to avoid to train a complex GAN, which leads to more instability during training. This aspect points to the limit of having an online bias-correction (at least for the medical data case).


**Experience Assessment:**

I have published one or two papers in this area.

**Review Assessment: Checking Correctness Of Derivations And Theory:**

I carefully checked the derivations and theory.

**Review Assessment: Checking Correctness Of Experiments:**

I carefully checked the experiments.

**Review Assessment: Thoroughness In Paper Reading:**

I read the paper thoroughly.

---

> ### Author Response · Authors · 2019-11-07
> **Response to AnonReviewer1**
>
> We thank the reviewer for the constructive and positive comments. Please see our detailed response to each of the points below.
>
> ————-
> >>> Comment:  Here are my main comments for this work:
> - Why Pearson’s correlation should be a reliable metric to quantify bias? This metric is insensitive to affine scaling of the data, which is a quite common form of bias (for example in medical images).
>
> > Response: In the revised manuscript, we added a rigorous proof of the statistical property of using Pearson’s correlation as the adversarial training objective (Section 3.1). We showed that the learned features and bias would achieve ‘mean independence’, a very strong type of statistical independence, thanks to the adversarial network being able to remove non-linear association (not just confined to affine) between variables.
>
> ————-
> >>> Comment: - The authors should have investigated the relationship between the proposed method and bias removal through canonical correlation analysis (CCA), and perhaps its non-linear variants. At the end this is what their network is doing, although in an end-to-end fashion. Using the CCA projections in the latent space for classification would be the closest approach to the state of the art for bias removal in statistical analysis (residual analysis).
>
> > Response: We have not figured out which specific state-of-the-art CCA bias-removal work the reviewer is referring to. In this work, we aim to equip the end-to-end convolution training with a rigorous way of removing bias/confounder. That was the main reason we did not consider applying traditional methods (linear or non-linear) to raw images, which are the inputs to deep learning and convolutional methods. However, we have also compared with a widely-used traditional method in all medical studies (Residualization) in Table 1. Our method outperforms it by a large margin.
>
> Note that the adversarial component essentially maps the learned features to the 1D space of bias, from which the correlation is computed. This echoes the concept of CCA, only in a non-linear manner. Therefore, our method is very similar to what the reviewer mentioned above. Please also note that we cannot directly use CCA in the context of convolutional networks unless we use its variations (such as Andrew et al. ICML 2013). In order to learn bias-invariant features we had to adversarially inject resilience into the convolutional part (Feature Extract, FE, component) in an end-to-end fashion. This can be a future research direction. We added this point in the Conclusion section, thanks to the reviewer.
>
> ————-
> >>> Comment: - The experimental setting illustrated in 4.1 is not clear. In which sense \sigma_A is a common factor for the two groups? Why the theoretical maximum classification accuracy is 90%? Figure 2 is not clear either and doesn’t help understanding the structure of the generated data (e.g. axis labels missing, colorbars units not specified).
>
> > Response: We assume the difference in \sigma_A (U(1,4) and U(3,6)) between the two groups represents true discriminative cues, and the difference in \sigma_B represents a confounding effect. We have reworded this in the text accordingly to avoid confusion.
>
> Since \sigma_A is the only discriminative cue in the dataset and is sampled from two overlapping intervals (U(1,4) and U(3,6)) for the two groups respectively, the theoretical maximum classification accuracy is 90%, i.e., 50% accuracy when \sigma_A is sampled from U(3,4) which is common between the two classes, and 100% accuracy for the remaining intervals. We have also clarified this point in the revised manuscript.
>
> The axis-label and coordinates of 2D t-SNE do not have any meaning, as it only shows relative local neighborhoods among instances. This is a very common practice for visualizing high dimensional data in lower dimensions in which the relative neighborhood of the high dimensional data is kept. As for the colorbar label in Figure 2, it was explained in the text that it is the value of \sigma_b, but according to this reviewer’s comment, we have added this information in the caption as well.

---

> > ### Author Response · Authors · 2019-11-07
> > **Response to AnonReviewer1 - Part 2**
> >
> > ————-
> > >>> Comment: - It is not clear why authors quantify the correlation in the latent space with tSNE projections. tSNE is highly sensitive to the choice of parameters and it would be important to ensure that it was a “fair competition” between all the methods, when showing the results of the dimensionality reduction. This is another modelling step relying on specific assumptions which decreases interpretability of the findings. The authors also proposed to assess the decorrelation of the estimated features throughout the different methods by measuring the squared distance correlation. Naturally, their method is the one which exhibits the best performances using this metric. However, this way of assessing the decorrelation of the features with the biases is unfair to the other methods, as in their case they specifically built their model to avoid statistical correlation between features and biases.
> >
> > > Response: Yes, “to avoid statistical dependence (linear or non-linear) between features and biases” is exactly what we want our model to have and what other models do not have. Note that we achieved this by adversially optimizing linear correlation between predicted and true bias, and we would want to validate that features and bias were independent while producing accurate classification. Neither tSNE nor distance correlation in the high-dimensional feature space is a “modelling step”; they are just ways to quantitatively (dist corr) and qualitatively (t-SNE) validate the “statistical independence” between features and biases after the training. We agree there are other visualization techniques and ways for measuring high-dimensional dependency, but we adopted them because they are the two most popular and relevant choices.
> >
> > Dist corr does not have hyperparameters, while in t-SNE the most important hyperparameter is the perplexity. All the compared methods reduced the images to the same dimensionality of the extracted feature vector, and we made sure that all these methods were evaluated with the same perplexity hyperparameter. This ensures exact same settings for all methods for a simple qualitative comparison.
> >
> > ————-
> > >>> Comment: - Experiment 4.2 has some controversial aspects, as the bias correction is performed on the control population only, while the model is trained on the entire population. I understand the fact that confounding can be estimated only on healthy conditions, however in this case the network is going to be biased by the control group by construction. The effect of such a choice in the end-to-end optimisation scheme is really not clear.
> >
> > > Response: As the reviewer acknowledged, modeling confounding effect within controls is the most acceptable practice in medical applications due to the fact that the confounding effect can be heavily distorted in diseased subjects. For many neuroscience studies based on traditional machine learning methods (Skelly et al. 2012; Adeli et al. 2018; Brookhart et al. 2010), it is also a standard practice to determine confounding effects only in controls and remove these effects for all subjects. In medical image analysis and neuroscience studies, the confounding factors are regressed out from the features by parameterizing a general linear model, GLM, (Madsen et al. 2010) on the controls of the training data. After parameterizing the GLM, the model is applied to the features of all samples in the dataset to residualize them, as done in (Rao et al., 2017; Adeli et al. 2018; Brookhart et al. 2010).
> >
> > In practice, we also observed slight accuracy drop in HIV classification if we performed confounder prediction for both groups. In the gender-shades experiment, however, the bias correction was performed on all subjects, since there are no distorted (diseased) groups in the study.
> >
> > (Adeli et al. 2018) Ehsan Adeli et al.  Chained regularization for identifying brain patterns specific to HIV infection. NeuroImage, 183:425–437, (2018).
> > (Brookhart et al. 2010) Brookhart, M. A., Stürmer, T., Glynn, R. J., Rassen, J., & Schneeweiss, S. (2010). Confounding control in healthcare database research: challenges and potential approaches. Medical care, 48(6 0), S114.
> > (Madsen et al. 2010) Madsen H, Thyregod P (2010): Introduction to general and generalized linear models. CRC Press.
> > (Rao et al. 2017) Anil Rao et al.  Predictive modelling using neuroimaging data in the presence of confounds. NeuroImage, 150:23–49, (2017).
> > (Skelly et al. 2012) Skelly, Andrea C., Joseph R. Dettori, and Erika D. Brodt. "Assessing bias: the importance of considering confounding." Evidence-based spine-care journal 3.01 (2012): 9-12.

---

> > > ### Author Response · Authors · 2019-11-07
> > > **Response to AnonReviewer1 - Part 3**
> > >
> > > ————-
> > > >>> Comment: - We also observe that the results of the baseline CNN are very close to the BR-Net. The main difference lies in the fact that the CNN tends to have an unbalanced classification between true negatives and true positives. However, what would happen if we corrected for age before applying the CNN ?
> > >
> > > > Response: As we have argued in the introduction, although confounder correction in traditional machine learning methods is simple and clear, it is not trivial at all in the nowadays end-to-end training of CNNs and often overlooked. There is no way of correcting confounders on raw images: “the raw intensities are only meaningful within a neighborhood but variant across images“. Removing effects from pixel values is incorrect. Raw intensity values may be different across different MR images taken by two different scanners, but what is important is how the intensities change in neighborhoods that define shapes and structures (in MRI studies white matter and gray matter areas). Therefore, it is impossible to correct for age as the input to the end-to-end training framework is the raw pixel (or voxel) intensity values. Because of this issue, considering the effects of biases and confounders in deep learning methods applied to medical images have been neglected. Our study shows a solution to this problem by adversarially removing the bias effects.
> > >
> > > Please note that we have already included in Table 1 a model that removes the effect of age from brain measurements (not intensity values) and applies a classifier after that (i.e., Resid+SVM). These measurements were extracted by a widely used medical imaging software (FreeSurfer) by extracting the volumes of brain predefined regions of interest. This also satisfies the reviewer’s comment of having a baseline that first regresses out the effect of age and then applies a classifier. Note that since the inputs are measurements after we remove the effect of age, we cannot apply CNN and therefore an SVM classifier is used.
> > >
> > > Additionally, as the reviewer noted, our main goal was to learn features that are statistically independent from the bias and confounder values, which is validated by various visualizations quantitatively and qualitatively. But please note that in many cases (e.g., in the HIV results of Table 1 and Gender-Shades results in Fig. 7 for Shade 6) our results are significantly better than all baselines.
> > >
> > > ————-
> > > >>> Comment: - In the case where the performances of the CNN would be improved, this last question would raise another one. Indeed, if I already know the confounding effects I want to correct for, why wouldn’t I correct them beforehand in order to avoid to train a complex GAN, which leads to more instability during training. This aspect points to the limit of having an online bias-correction (at least for the medical data case).
> > >
> > > > Response: We appreciate this comment from the reviewer. But as also mentioned above, there is no way of correcting confounders on raw images as the raw intensities are only meaningful within a neighborhood but variant across images. Removing effects from pixel values is incorrect. Raw intensity values may be different across different MR images taken by two different scanners, but what is important is how the intensities change in neighborhoods that define shapes and structures (in MRI studies white matter and gray matter areas). Therefore, it is impossible to correct for confounders as the input to the end-to-end training framework is the raw pixel (or voxel) intensity values. Because of this issue, considering the effects of biases and confounders in deep learning methods applied to medical images have been long neglected.  Removing the effect of confounders in traditional machine learning with engineered features was a common practice that is not doable in end-to-end deep learning frameworks. Our study shows a solution for this problem to adversarially remove the bias effects.

---

### Official Review · AnonReviewer2 · 2019-10-24
**Official Blind Review #2**

**Rating:** 3

**Review:**

This paper proposes an adversarial approach toward debiasing neural network representations w.r.t protected attributes. The core idea is balance task loss with an adversarial loss from which protected attributes have low correlation with the feature representation used for the end task. The paper provides some synthetic experiments, and evaluates on HIV data (the bias variable being age) and gender classification in images, stratified by skin shade (the bias variable being skin shade). Results demonstrate improved balanced accuracy across all three experiments.

Overall the direction is interesting and the methodology is intuitive and sound. Unfortunately, this paper seems unaware of extremely related works:

1. (AAAI 2018) http://www.m-mitchell.com/papers/Adversarial_Bias_Mitigation.pdf
2. (ACL 2018) https://arxiv.org/abs/1808.06640
3. (ICCV 2019) https://arxiv.org/abs/1811.08489
4. (ICCV 2019) http://hal.cse.msu.edu/assets/pdfs/papers/2019-iccv-kernel-adversarial-representation-learning.pdf

Beyond experiments on different datasets, and slight modification of the adversarial loss for correlation, I am unsure what this paper contributes beyond these works. While some some of these papers can potentially be considered contemporary, authors must at least address these issues.  Furthermore, modification of the objective seems to have mixed results (Table 2 CatGAN vs. BR-NET, although I'm not sure whats going on with vgg vs resnet), where the baseline would correspond more closely to the setup in https://arxiv.org/abs/1811.08489 (3. above) .

Overall I am positive about the direction, but I am unsure this paper represents a significant contribution over existing work.



**Experience Assessment:**

I have published in this field for several years.

**Review Assessment: Checking Correctness Of Derivations And Theory:**

I assessed the sensibility of the derivations and theory.

**Review Assessment: Checking Correctness Of Experiments:**

I carefully checked the experiments.

**Review Assessment: Thoroughness In Paper Reading:**

I read the paper thoroughly.

---

> ### Author Response · Authors · 2019-11-07
> **Response to AnonReviewer02**
>
> We thank the reviewer for the constructive and positive comments. Please see our detailed response to each of the points below.
>
> ————-
> >>> Comment: Overall the direction is interesting and the methodology is intuitive and sound. Unfortunately, this paper seems unaware of extremely related works:
> 1. (AAAI 2018) http://www.m-mitchell.com/papers/Adversarial_Bias_Mitigation.pdf
> 2. (ACL 2018) https://arxiv.org/abs/1808.06640
> 3. (ICCV 2019) https://arxiv.org/abs/1811.08489
> 4. (ICCV 2019) http://hal.cse.msu.edu/assets/pdfs/papers/2019-iccv-kernel-adversarial-representation-learning.pdf
>
> > Response: Thank you for pointing us to these prior works. We have added them in literature review accordingly and have discussed their differences to our work. Please note that we have already cited some of these articles (e.g., Xieet al. NeurIPS 2017 and Akuzawa et al. EMCL 2019). As we were already aware of several of these works, we regarded our work quite distinct from existing state-of-the-art in the sense that our work could handle continuous bias/confounder variables as opposed to many works focusing only on binary/categorical variables. Please also consider that two of the above works (3 and 4) are ICCV 2019 papers, which were published online on arxiv and CVF OpenAccess in October 2019, which was after the deadline of ICLR submission.
>
> As opposed to other works, we also explicitly modeled the statistical dependency between features and bias. To make clear the difference from prior arts, please refer to Section 3.1 in the updated manuscript describing the statistical property of our method. We theoretically showed that modifying the adversarial loss as the ‘Pearson correlation’ actually guarantees an important statistical property of the model: the learned features and bias would achieve ‘mean independence’, a much stronger type of statistical independence than linear independence (Pearson correlation=0). Such property can only be achieved by our strategy of adversarially optimizing the Pearson-correlation of predicted and true bias. This distinguishes our work from several others, in which adversarial object is solely defined w.r.t. maximizing bias prediction loss (only aims at predicting the exact value of the loss neglecting proper statistical (in)dependence).
>
> ————-
> >>> Comment: Beyond experiments on different datasets, and slight modification of the adversarial loss for correlation, I am unsure what this paper contributes beyond these works. While some of these papers can potentially be considered contemporary, authors must at least address these issues. Furthermore, modification of the objective seems to have mixed results (Table 2 CatGAN vs. BR-NET, although I'm not sure whats going on with vgg vs resnet), where the baseline would correspond more closely to the setup in https://arxiv.org/abs/1811.08489 (3. above).
>
> > Response: As for the novelties of the paper, please see the above text. Our experiments show that strategies taken by other methods do not work for continuous and ordinal variables. These types of bias/confounder variables are very common in real-world computer vision and medical applications.
> CatGan (the same idea of the baseline suggested by the reviewer) also worked reasonably well with VGG backbone but was only applicable when bias took discrete categorical values. Even in that case, the results are BR-Net are significantly superior for some shade categories (shade 6, see Fig. 7). Please note that in the HIV experiment CatGan is not applicable as the bias variable (age) is a continuous variable.
>
> ————-
> >>> Comment: Overall I am positive about the direction, but I am unsure this paper represents a significant contribution over existing work.
>
> > Response: Thank you very much for the positive and constructive feedback. Please also refer to the repnoses above. In summary, we would like to again mention that our proposed method is not just proposing a new loss function. We explicitly modeled the statistical dependency between features and bias. We added theoretical properties of the proposed method in the updated manuscript (Section 3.1). We theoretically showed that using ‘Pearson correlation’ as the adversarial loss actually guarantees that the learned features and bias would achieve ‘mean independence.’ Such property can only be achieved by our strategy of adversarially optimizing the Pearson-correlation of predicted and true bias. This distinguishes our work from the previous work. The experimental evaluations are also designed to show these claims and that is why we selected [BR-Net with categorical adversarial loss (CatGAN)], [BR-Net with MSE adversarial loss], and [Multi-Task] and direct baselines for our method. This way, we have directly implemented and compared with all previous strategies (including 1, 2, 3, and 4 referred to by the reviewer above).

---

> > ### Public Comment · ~Vishnu_Boddeti1 · 2019-11-07
> > **More missing related work on Adversarial Representation Learning**
> >
> > This paper misses more related work on this topic, including papers that address the same problem and propose very similar solutions.
> >
> > 1) Kim, B., Kim, H., Kim, K., Kim, S., & Kim, J. "Learning Not to Learn: Training Deep Neural Networks with Biased Data." CVPR 2019 ---- This paper uses adversarial training similar to Xie et al. NeurIPS 2017.
> >
> > 2) Roy, P., & Boddeti, V. Mitigating Information Leakage in Image Representations: A Maximum Entropy Approach. CVPR 2019 ----- This paper shows that gradient reversal based adversarial training is ill-posed with oscillatory behavior. The paper proposes a non-zero sum game instead that stabilizes the optimization in theory and to a large extent in practice.
> >
> > 3) Madras, D., Creager, E., Pitassi, T., & Zemel, R. "Learning adversarially fair and transferable representations," ICML 2019 ----- This paper considers loss functions for the adversary that are not cross-entropy or MSE. Instead it considers loss functions corresponding to group fairness notions including demographic parity, equalized odds and equal opportunity.
> >
> > The next group of papers look at this problem from a mutual information perspective.
> >
> > 1) Song, J., Kalluri, P., Grover, A., Zhao, S., & Ermon, S. "Learning Controllable Fair Representations," AISTATS 2019
> >
> > 2) Bertran, M., Martinez, N., Papadaki, A., Qiu, Q., Rodrigues, M., Reeves, G., & Sapiro, G. "Adversarially Learned Representations for Information Obfuscation and Inference," ICML 2019
> >
> > 3) Moyer, D., Gao, S., Brekelmans, R., Galstyan, A., & Ver Steeg, G. "Invariant Representations without Adversarial Training," NeurIPS 2018 ---- This paper proposes a mutual information based solution without an explicit adversary.
> >
> > Lastly, the Sadeghi et al, ICCV 2019 paper minimizes the minimum MSE between b and \hat{b} with the adversary being a linear regressor. This is exactly the same as maximizing the Pearson Correlation between b and \hat{b}. See Lemma 1 and Lemma 4 of the arxiv version.
> >
> > I am going to stop here but there are more at the link below in case anyone is interested.
> > http://hal.cse.msu.edu/blog/adversarial-representation-learning/

---

> > > ### Author Response · Authors · 2019-11-08
> > > **Response to the comment form Dr. Vishnu Boddeti**
> > >
> > > > Response: Thank you for listing additional relevant works. We acknowledge that there are a body of recent works on invariant feature learning, but as we have stated, we believe our work is the first to consider the statistical (mean) independence between features and bias by using Pearson correlation as the adversarial objective in a minimax adversarial optimization, which can successfully remove bias as a continuous variable and ensure statistical independence of features from the bias. Please note that we emphasized that we are aware of the large body of research this year on this topic and we cited some of them. This high number of quality publications on this topic itself shows its importance. But we argued that using simple and widely used loss functions (MSE or cross-entropy) as used in all these works is not the best choice to remove the effects of all types of biases. We proposed a loss that removes the statistical dependence, backed up with theoretical analysis, and is usable in adversarial techniques.
> > >
> > > We emphasize again that even though bias prediction for continuous variables can be achieved by both minimizing MSE (loss of regressor) and maximizing correlation, MSE _cannot_ be used as the adversarial loss in the minimax optimization; while minimizing correlation (magnitude) removes statistical dependence, maximizing MSE is an ill-posed objective. We have stated in the paper “In fact, the adversarial training based on MSE leads to the maximization of the L2 distance between bˆ and b, which could be trivially achieved by uniformly shifting the magnitude of bˆ, thereby potentially resulting in an ill-posed optimization and oscillation in the adversarial training.” We have also shown this argument in all three experiments that MSE is a suboptimal metric for minimax optimization.
> > >
> > > As for the specific work mentioned here (your paper, Sadeghi et al. ICCV 2019, the arxiv version appeared only two weeks ago after ICLR submission deadline), we find the theoretical interpretation really strong and appealing, but the framework is not based on the minimax adversarial training strategy, and most theoretical results (lemmas) are only valid for a limited family of networks (“one-layer” linear regressors only), whereas our work is targeted at generic networks including ConvNets and multi-layer perceptrons. Going beyond linear methods is the gist of modern deep learning that learns features in an end-to-end way. Please note that our formulation is not just a linear correlation removal. Section 3.1 explicitly proves that although we used Pearson correlation as the loss function, the multi-layer adversarial component removes higher order statistical differences between features and the bias (very similar to the case that one uses a linear or logistic regression loss function in a multi-layer network and can solve nonlinear problems).

---

### Author Response · Authors · 2019-11-07
**General feedback on the initial reviewer comments**

We thank all the reviewers for their invaluable and constructive feedback. We appreciate all the reviewers noting that this paper is interesting and addresses an important problem. According to the reviewers’ feedback, we have uploaded a revision to our paper, addressing the comments.

The novelty of our paper is the proposal of a new strategy to remove statistical dependence of the learned features to any bias (or confounder) existing in the dataset or the problem setup. As opposed to the previous works for adversarial bias-removal, we explicitly modeled the statistical dependency between features and bias. To make clear the difference from prior arts, we added Section 3.1 in the updated manuscript describing the statistical property of our method. We theoretically showed that modifying the adversarial loss as the ‘Pearson correlation’ actually guarantees an important statistical property of the model: the learned features and bias would achieve ‘mean independence’, a much stronger type of statistical independence than linear independence (Pearson correlation=0). Such property can only be achieved by our strategy of adversarially optimizing the Pearson-correlation of predicted and true bias. This distinguishes our work from the previous works, in which adversarial object is solely defined w.r.t. maximizing bias prediction loss (only aims at predicting the exact value of the loss neglecting proper statistical (in)dependence). The experimental evaluations were also designed to show these claims and that is why we selected [BR-Net with categorical adversarial loss (CatGAN)], [BR-Net with MSE adversarial loss], and [Multi-Task] as direct baselines for our method. This way, we have directly implemented and compared with the strategies used by the previous work.

Another aspect of our work is that it explores bias in a wide range of applications: from fairness in machine learning to confounding effects in medical studies. This creates an umbrella concept that our method can be applied to a wide range of applications. Specifically for the medical application, in Section 4.2, we mentioned that the bias (confounder) adversarial loss can only be back-propagated to the feature extraction (FE) component when applied to only control subjects. This is a well-known concept in medical studies when dealing with confounders using traditional statistical methods (stratification or residualization), which was not directly applicable to deep learning approaches.

---

### Author Response · Authors · 2019-11-14
**Towards the end of the discussion period**

Dear Reviewers,

Since we have updated our paper with a revision addressing all comments and provided a point-to-point response over a week ago, it could be very nice to hear the feedback from the reviewers while there is still time to respond. Thanks again for taking the time to review the paper and for all the constructive feedback.

---

### Decision · Program_Chairs · 2019-12-19

**Decision:**

Reject

**Comment:**

The paper addressed the problem of machine bias when training machine learning models. The authors propose an approach based on representation learning with adversarial training. As opposed to the majority of previous works that trying to create a representation from which it is not possible to predict the sensitive feature (bias), the authors propose to minimize the dependency between the learned features and the sensitive feature with adversarial training. While acknowledging that the proposed model is addressing an important problem and is potentially useful, the reviewers and AC note the following potential weaknesses:
(1) limited technical contribution -- the proposed approach is similar to a number of works published in machine learning and computer vision before the submission deadline that were overlooked by the authors. Specifically: i) adversarial training for learning fair representations [Edwards and Storkey, Censoring Representations with an Adversary, ICLR 2016], [Beutel, et al 2017, Data decisions and theoretical implications when adversarially learning fair representations], ii) learning fair representation by minimizing the dependency between the latent representation and the sensitive attributes [The variational fair autoencoder, ICLR 2016 by Louizos et al.; Fairness Constraints: Mechanisms for Fair Classification, by Zafar et al, 2015] or by minimizing the mutual information between feature embedding and bias [Learning Not to Learn: Training Deep Neural Networks with Biased Data, CVPR 2019].
(2) Limited empirical evidence -- the baseline methods used in the evaluation are not sufficient to assess the benefits of the proposed approach over the existing SOTA methods mentioned above. In fact, none of the baseline methods used in the evaluation tackle machine bias (via adversarial training or minimizing statistical dependence).
(3) It would be beneficial to also report fairness metrics, e.g. equality of opportunity, statistical parity, to assess the effectiveness of bias removal. R1 has raised some concerns regarding empirical evidence -- see the point about mixed results. Also R2 has reported concerns regarding controversial results in experiment 4.2 and suggested ways to justify when and why the results of the CNNs baseline are close to the BR-Net. Addressing these concerns would strengthen the contributions of the proposed method.

Among these, (3) did not have a decisive impact on the decision, but would be helpful to address in a subsequent revision. However, (1) and (2) make it very difficult to assess the benefits of the proposed approach, and were viewed by AC as critical issues. AC suggests, in its current state the manuscript is not ready for a publication. We hope the reviews are useful for improving and revising the paper.